# Nucleus Near-Infrared (nNIR) Irradiation of Single A549 Cells Induces DNA Damage and Activates EGFR Leading to Mitochondrial Fission

**DOI:** 10.3390/cells11040624

**Published:** 2022-02-11

**Authors:** Momoh Gbetuwa, Long-Sheng Lu, Tsung-Jen Wang, Yin-Ju Chen, Jeng-Fong Chiou, Tai-Yuan Su, Tzu-Sen Yang

**Affiliations:** 1Graduate Institute of Biomedical Materials and Tissue Engineering, Taipei Medical University, Taipei 110, Taiwan; d825107004@tmu.edu.tw (M.G.); lslu@tmu.edu.tw (L.-S.L.); yjchen1113@tmu.edu.tw (Y.-J.C.); 2International PhD Program in Biomedical Engineering, Taipei Medical University, Taipei 110, Taiwan; 3Department of Radiation Oncology, Taipei Medical University Hospital, Taipei Medical University, Taipei 110, Taiwan; solomanc@tmu.edu.tw; 4Department of Medical Research, Taipei Medical University Hospital, Taipei 110, Taiwan; 5TMU Research Center of Cancer Translational Medicine, Taipei Medical University, Taipei 110, Taiwan; 6Center for Cell Therapy, Taipei Medical University Hospital, Taipei Medical University, Taipei 110, Taiwan; 7International PhD Program for Cell Therapy and Regeneration, Taipei Medical University, Taipei 110, Taiwan; 8Department of Ophthalmology, Taipei Medical University Hospital, Taipei 110, Taiwan; tjw@tmu.edu.tw; 9Department of Ophthalmology, School of Medicine, College of Medicine, Taipei Medical University, Taipei 110, Taiwan; 10Department of Radiology, School of Medicine, Taipei Medical University, Taipei 110, Taiwan; 11Department of Electrical Engineering, Yuan-Ze University, Chung-Li 32003, Taiwan; tysu@saturn.yzu.edu.tw; 12Graduate Institute of Biomedical Optomechatronics, Taipei Medical University, Taipei 110, Taiwan; 13School of Dental Technology, Taipei Medical University, Taipei 110, Taiwan; 14Research Center of Biomedical Device, Taipei Medical University, Taipei 110, Taiwan

**Keywords:** near infrared (NIR), epidermal growth factor receptor (EGFR), mitochondrial fragmentation count (MFC), mitochondrial dynamic, cetuximab, caffeine, PD153035

## Abstract

There has been great interest in identifying the biological substrate for light-cell interaction and their relations to cancer treatment. In this study, a near-infrared (NIR) laser is focused into the nucleus (nNIR) or cytoplasm (cNIR) of a single living cell by a high numerical aperture condenser to dissect the novel role of cell nucleus in mediating NIR effects on mitochondrial dynamics of A549 non-small cell lung cancer cells. Our analysis showed that nNIR, but not cNIR, triggered mitochondrial fission in 10 min. In contrast, the fission/fusion balance of mitochondria directly exposed to cNIR does not change. While the same phenomenon is also triggered by single molecular interactions between epidermal growth factor (EGF) and its receptor EGFR, pharmacological studies with cetuximab, PD153035, and caffeine suggest EGF signaling crosstalk to DNA damaging response to mediate rapid mitochondrial fission as a result of nNIR irradiation. These results suggest that nuclear DNA integrity is a novel biological target for cellular response to NIR.

## 1. Introduction

Current cancer therapy is composed of three major treatment modalities, including surgery, radiation, and chemotherapy [1]. There is a great interest to improve these modalities or to introduce new modalities to increase therapeutic index and to maximize potential benefits and to minimize the associated risk [2]. Phototherapy is one of such promising modality due to its excellent safety profile and successful preclinical and clinical experiences [2,3]. Near infrared (NIR) light is a reasonable light source for clinical phototherapy due to its compatibility with biological tissues, accuracy/sensitivity, and deep penetration into the tissue (>1 cm) [2,3]. Phototherapy is one such promising modality due to its excellent safety profile and successful early clinical experiences. While common cancer therapeutics target subcellular compartments such as the nucleus [4,5,6], mitochondria [5,7] and the plasma membrane [8,9,10] to provide anticancer efficiency, the subcellular target for phototherapy remains poorly characterized. NIR light is a reasonable light source for clinical phototherapy due to its compatibility with biological tissues [11,12]. A deeper understanding of the biological basis of subcellular response to NIR will be beneficial to inspire the design of next generation NIR phototherapy. In order to improve the NIR phototherapy and avoid over-dosing related complications, it is very important to understand the biological basis of cellular response to NIR [13]. NIR treatment of cells have a linear relationship (that is, the higher the laser doses, the higher the percentage of cell death). However, this linear relationship has its drawbacks, such as the absence of providing an appropriate or maximum energy dose can affect the therapeutic applications and the higher the laser dose, the higher the harmful effects on the target cells. Nevertheless, studies have shown that the highest percentage of cell death can be achieved at 299,559 J/cm^2^ laser dose [14]. In this study, we therefore treated cells with 224.02 J/cm^2^ NIR to achieve a better clinical outcome and to limit the drawbacks of longer exposure to NIR laser light.

Mitochondria are an attractive subcellular target to mediate cell response to NIR. The organelle is the major depot of extranuclear DNA and is responsible for adenosine triphosphate (ATP) synthesis via the electron transport chain. The complex IV in the electron transport chain, responsible for transferring electrons to molecular oxygen and creating electromotive force across the mitochondrial inner membrane, contains cytochrome c that absorbs NIR, and leads to loss of mitochondrial membrane potential [15]. In addition to energy generation, mitochondria are an organelle hub that integrate environmental signaling inputs. These signals frequently come from growth factor engagement of corresponding receptors on the cell surface, such as epithelial growth factor (EGF) binding to its receptor (EGFR) or activated kinase cascade after deoxyribonucleic acid (DNA) damage as an intrinsic DNA damage response (DDR) [16,17]. Furthermore, mitochondria dynamically alter their structure between fusion and fission states in response to these stimuli and crosstalk to the nuclear transcription machinery to orchestrate key cellular fate decisions, including live, death, and stress adaptation [18,19,20]. Although it has been long known that NIR interferes with mitochondrial signal transduction [21], it remains uncertain about the mechanistic roles by NIR absorption by mitochondrial cytochromes and/or extra-mitochondrial cellular signals.

Epidermal growth factor receptor (EGFR) has been one of the most effective oncogenes that are usually altered in cancers [22]. Inhibitors targeting this pathway significantly improve the clinical outcome in patients with solid tumors that are associated with EGFR mutations [23,24] or gene amplification/overexpression of EGFR proteins [25]. There are seven EGFR ligands that has been described that have shown to induce specific cellular response and intracellular trafficking events that occur in both in vitro and in vivo [26]. These different signaling properties of the various ligands have been related to their ability to differentially stabilize the EGFR dimers which determines the specific signaling output [27]. Targeting overexpressed EGFR proteins with cetuximab enhances mitochondrial-triggered apoptosis and is an effective treatment for EGFR overexpressing tumors [28]. Inhibiting EGFR tyrosine kinase activities with tyrosine kinase inhibitor (such as PD153035) is another approach to control excessive EGFR signaling [29]. Despite its proven importance in clinical oncology, it remains unknown if EGFR signaling is involved in photobiology of cancer cells and therefore be a potential photosensitizing pathway for novel therapeutic development.

Nanodiamonds (NDs) have special carbon nanoparticles that have gained attention for their biocompatibility, high functional surfaces, and their optical and physical properties. These properties have provided a good regenerative platform for medicine that provides an application that ranges from targeted delivery of drugs [30]. They are said to be chemically inert and have a very small cytotoxicity in vivo [31,32,33]. In drug delivery, one major concern is the toxicity of nanomaterials therefore the material with non-toxic and that is biocompatible is highly recommended in clinical use. Other studies have used fluorescence nanodiamond (FND) and cetuximab to target EGFR that expresses cancer cell to deliver drugs [34,35]. We used FND in this study for the targeting of drug delivery approaches and use FND surfaces attached with different ligands such as EGF for the reorganization of EGFR receptor to enhance endocytosis of EGF into the cytosol domain of a cell to facilitate drug such as cetuximab, caffeine, and PD153035 reactivity.

In this study, we examined the effect of NIR on EGFR activities in the presence of treated cells with drugs such as PD153035, caffeine, and cetuximab. NIR treatment in subcellular organelles is very crucial to accurately focus light on the targeted subcellular target of interest. In order to investigate the subcellular substrate for NIR-mitochondrial interaction, our group took a single cell photomodulation approach to explore the role of organelle-directed NIR irradiation. With a single cell photomodulation platform composed of an 830 nm infrared diode laser, an electrical shutter, laser-focusing optics (condenser), we specifically delivered sublethal levels of NIR laser to either cytosolic mitochondria (cNIR) or nuclei (nNIR) of an A549 cell, which is derived from non-small cell lung cancer with wild type EGFR that only activates the signal upon stoichiometrically matched EGF engagement. First, we quantitatively monitored mitochondrial dynamic change towards fission state as a function of time, which serves as an immediate mitochondrial response to organelle-directed NIR. Then we took advantage of the pharmacological blockade to dissect the relative contribution of EGFR signaling and DDR to immediate mitochondrial response to cNIR and nNIR. Finally, we applied the mitochondrial fragmentation count (MFC) to quantify the mitochondrial pattern and mitochondrial dynamics. Our results show that NIR modulation of mitochondrial dynamics is mainly regulated by cellular signalings, such as EGFR and DDR, rather than by events triggered within the mitochondria.

## 2. Materials and Methods

### 2.1. Reagents and Materials

A549 (adenocarcinoma cells) cell line, Dulbecco’s Modified Eagle’s Medium—high glucose (DMEM) power, Trypsin and 10× Phosphate buffer saline (PBS) Sterile, Carbonyl cyanide m-chlorophenyl hydrazone (CCCP) and Mitochondrial division inhibitor 1 (mdivi-1) were ordered from Merck Life Science, Science, Dorset, UK Limited. Cetuximab was bought from Medical Store (MS) Asanwa Ahmedabab Gujarat, India. Caffeine was ordered from CSPC Pharmaceutical Group Limited, Shijiazhuang, China. MitoTracker Green was bought from Thermo Fisher Scientific, Loughborough, UK. Fluorescence Nanodiamond (FND) was bought from FND BIOTECH, Inc. Taipei, Taiwan. PD153035 was bought from MedChemExpress LLC (MCE), Princeton, NJ, USA. Biotin EGF was bought from Molecular Probes, Invitrogen, Carlsbad, CA, USA. Both MitoTracker Green and mdivi-1 were dissolved as stock solutions in dimethyl sulfoxide (DMSO) for dilution in complete media.

### 2.2. Cell Culture and Drug Treatments 

A549 cells were cultured in Dulbecco’s modified Eagle’s medium (DMEM) supplemented with 10% fetal bovine serum at 37 °C in a humidified incubator with 95% air and 5% CO_2_ to allow for appreciable confluence. To promote adhesion, cultured cells were seeded on a collagen one-coated glass coverslip microchannel for 22 h at 37 °C to ensure 60–70% cell confluence was achieved. Cultured cells on glass coverslip were then assembled onto the bottom of the flow chamber and treated with 0.3 µM MitoTracker Green and incubated for 30 min at room temperature then washed with PBS.

### 2.3. Near Infrared (NIR) Laser Alignment and Florescence Imaging

The experimental setup for the single-cell NIR laser irradiation system is shown in Figure 1. An 830 nm (Lambda Beam Wavelock, RGB Photonics, Bavaria, Germany) laser beam was passed through a plano-convex lens L1 (focal length = 25 mm) and L2 (focal length = 100 mm). The measured output beam diameter was four times that of the input beam. This 830 nm laser beam then passed through a plano-convex lens L3 (focal length = 125 mm) and L4 (focal length = 250 mm) this makes the total output beam diameter twice that of the input beam thus this laser beam will expand eight times. The laser beam was then reflected by the dichroic mirror D1 (780dcspxr, Chroma Irvine, CA, USA) on the inverted microscope (TE2000U, Nikon, Tokyo, Japan) and passes through the condenser for focus in the sample contained microchannel below, where the laser spot area was 52.36 µm^2^. The back aperture of the condenser was conjugated with mirror M4 and L1 to be able to adjust the position of laser beam in microchannel. Images were obtained by the inverted microscope equipped with an objective lens (Plan Apo 60x/1.40 oil, Nikon, Tokyo, Japan), band-pass filters for a sCMOS camera (ORCA-Flash4.0 V3, Hamamatsu, Japan) and the fluorescence images from cells were acquired with a spatial resolution of image size, 572 × 332 pixels (the smallest unit of a digital image) with temporal resolutions of 500 ms and fluorescence green excitation wavelength 485 nm an emission of 530 nm to stain and analyze the mitochondrial structure.

### 2.4. Single Cell NIR Laser Light Treatment on A549 Cell

Cultured cells from protocol in Section 2.2. Two cells in the same image plane where one cell was exposed to nucleus 224.02 J/cm^2^ NIR (nNIR), for 10 s and the other cell not exposed to NIR the cells were then imaged at different time points. Single cell nucleus and cytosol were also exposed to 224.02 J/cm^2^ NIR for 10 s then imaged cells at 0 min (Before NIR), 1, 5, 10, 15, and 20 min and used ImageJ Software to analyze the mitochondrial fragmentation count (MFC), the laser spot is indicated as a small brown circle located either in the nucleus (nNIR) of single cell or in the cytosol (cNIR). In this study, each single cell is being copied five times. That is, for every single cell image there are five images. Therefore, in this study, for a single cell, there were 5 images; for 50 cells there were 250 images; and for 110 cells there were 550 images. The mean and standard error of the mean was therefore determined based on the number of images per cell.

### 2.5. Carbonyl Cyanide M-Chlorophenyl Hydrazone (CCCP) and Mitochondrial Division Inhibitor 1 (Mdivi-1) Treatment of A549 Cells and Single Cell Exposed to Nuclear NIR (nNIR) in the Same Image Plane

Here, we followed the method in Section 2.2 protocol. Prepare 20 μM CCCP reconstituted in DMEM without sodium pyruvate (-sp) and dispense into the volumetric flow system and incubate cells for 30 min at 37 °C, then washed with 20 μM CCCP reconstituted in PBS. We then treated cell with 20 µM mdivi-1 reconstituted in PBS and incubate for 30 min at 37 °C, then wash cells with 20 µM mdivi-1 reconstituted in PBS and imaged 110 single cell each. For the nuclear exposed NIR (nNIR) and the cytosol exposed NIR (cNIR) cells were exposed to nNIR and cNIR for 10 s the shutter was then closed, and the cells were imaged from 1, 5, 10, 15, and 20 min.

### 2.6. Treatment of A549 Cells with 1 µM PD153035, FND, 100 nM Cetuximab, 1 mM Caffeine and Conjugated 100 nM FND-EGF

Culture cells were prepared as in the protocol outlined in Section 2.2. We prepared conjugated 100 nM FND and incubated the cell for 22 h at 37 °C, the following day we treated the cells again with 1 µM PD153035 and conjugated 100 nM FND-EGF, 100 nM cetuximab and conjugated 100 nM FND-EGF and 1 mM caffeine conjugated 100 nM FND-EGF individually, incubated cells for 1 h at 37 °C through an inlet port of the flow chamber using an automated syringe pump, incubated cells in a dark with 0.3 µM MitoTracker Green reconstituted in PBS at 37 °C for 30 min. Washed cells with 0.3 µM MitoTracker Green, 1 µM PD153035 and conjugated 100 nM FND-EGF, 100 nM cetuximab and conjugated 100 nM FND-EGF and 1 mM caffeine conjugated 100 nM FND-EGF all been reconstituted in PBS individually and imaged individually drug treated cell (Before NIR). Expose the nucleus to 224.02 J/cm^2^ NIR for 10 s close laser and image (50 cells) cells at different time points of 10, 20, 30, 40, and 50 min for the individual drug treated cells. ImageJ Software was used to analyze the MFC. Due to the different experimental conditions in Section 2.5, different time points were used in this study, such as: (1) mdivi-1 and CCCP, incubated for 30 min because the drug depreciates over the 30 min incubation period; (2) MitoTracker Green, incubated for 30 min because prolonged incubation of MitoTracker Green leads to depreciation of the fluorescent dye; (3) A549 cells were exposed to 224.02 J/cm^2^ of NIR for 10 s, as exposure of cells to NIR laser for too long may result in severe tissue damage that will have little or no clinical benefit; and (4) cells were treated with 100 nM FND and incubated for 22 h. The next day, cells were treated with 1 μM PD153035 and conjugated with 100 nM FND-EGF and incubated for 1 h, as we realized in this study when cells were treated with 1 μM PD153035, conjugated with 100 nM FND-EGF and incubated for 1 h. The next day, cells were treated with 1 μM PD153035, conjugated with 100 nM FND-EGF and incubated for 1 h. We therefore designed another experimental protocol by first treating the cells with 100 nM FND and incubating these cells for 22 h. Then the cells were treated again with 1 µM PD153035 and conjugated with 100 nM FND-EGF and incubated for 1 h. We observed an increase in the number of FND on the cell membrane, which is shown in Appendix A. The same experimental protocol was applied to cetuximab and caffeine.

### 2.7. Mitochondrial Fragmentation Count (MFC)

MFC is the counting of mitochondrial particle and dividing it by the pixels of mitochondrial network. Using imageJ software, a summary of the average mitochondrial particle count was carried out, and mitochondrial fragmentation count was carried out by calculating the average particle count multiplied by 10,000 on the bases of random choice and then divided by the total mitochondrial pixel [36,37]. In this protocol of calculating the MFC, a single numerical value to each cell was assigned.

### 2.8. Statistical Analysis

Statistical comparisons were performed by one-way ANOVA followed by the Newman Keuls post-hoc test (GraphPad Prism software); a value of *p* < 0.05 was considered to indicate a statistically significant difference. All data were presented as the mean ± standard error of the mean (sem).

## 3. Results

### 3.1. Single Cell Exposed to Nuclear NIR (nNIR) in the Same Image Plane and Treatment of A549 Cells with Mdivi-1 and CCCP

To demonstrate that the single-cell NIR laser irradiation system can precisely control nNIR at the single-cell level without affecting the nearby cell, a single cell nucleus was exposed to NIR in the same image plane with another cell that was not treated with NIR this is shown in Figure 2A. These cells were imaged from 1, 5, 10, 15, and 20 min, and five mitochondrial fluorescence images were acquired at each observation time point for each single living cell. Therefore, the mean ± sem of MFC as a function of time can be determined.

Analysis of data showed a significant difference between the mitochondrial patterns (mitochondrial fission) before and after nNIR treatment, as it can be seen in the cropped area of Figure 2A1. At the same time point, the other cell not exposed to NIR showed the similar fine interwoven elongated mitochondrial structure, as shown in Figure 2A2. In addition, there was a significant (*p* < 0.01) increase in MFC (Cell-A1: MFC = 3.7 ± 0.2) for nuclear exposed cell but no MFC change (Cell-A2: MFC = 1.9 ± 0.2) in non-exposed NIR cell, as shown in Figure 2B. 

MFC of cells (*n* = 110) treated with mdivi-1 showed an elongated well defined mitochondrial structure but CCCP treated cells showed a fragmented and granular mitochondrial structure this is shown in Figure 3A. Analyzed MFC’s of cells treated with mdivi-1 and CCCP were compared, cells treated with 20 µM mdivi-1 was shown to have a significant (*p* < 0.01) decrease in MFC (1.5 ± 0.2) but analysis of cells treated with 20 µM CCCP showed a very significant (*** *p* < 0.01) increase in MFC (4.3 ± 0.2) when compared to the control as shown in Figure 3B.

### 3.2. A549 Cell Nuclear and Cytosolic Exposure to 224.02 J/cm^2^ NIR

Figure 4A illustrates the differences between mitochondrial dynamics of single A549 cells before 224.02 J/cm^2^ nNIR and after nNIR for 20 min. The cropped area before and after nNIR illustrating a significant difference between the mitochondrial patterns (mitochondrial fission) and the MFC before and after nNIR treatment. Figure 4B illustrates the differences between mitochondrial dynamics of single A549 cell before 224.02 J/cm^2^ cNIR and after cNIR for 20 min. The cropped area before and after cNIR illustrating no significant change in both the mitochondrial patterns and the MFC before and after cNIR treatment. There was a very significant (*** *p* < 0.01) gradual increase of MFC after 5 min in single cell nNIR but no change in the MFC in single cell cNIR as shown in Figure 4C. We again exposed 110 single cells to 224.02 J/cm^2^ nNIR and cNIR, and results showed a significant increase in the MFC of nNIR exposed cells, there was a sharp increase in MFC after 5 min to 15 min, MFC was then stable after 15 min but no significant change in the MFC of cNIR exposed cells Figure 4D. This finding demonstrates that single-cell measurements can identify the individual cellular responses of NIR irradiation and extend to calculate ensemble average when the number of single-cell samples increases.

### 3.3. A549 Cells Treated with FNDs, Free EGF, and 100 nM Conjugated FND-EGF

In this study, the fluorescent nanodiamond (FNDs) were used as a photostable fluorescent probe for the uptake of FND-EGF by A549 cells. When A549 cells were treated with 100 nM FND, there was indeed a small amount of non-specific uptake of FND in the cytosol of the cell (Appendix A). When A549 cells treated with FND for 22 h and further treated with 100 nM conjugated FND-EGF and incubated for 1 h, the fluorescence images showed very large amount of uptake of FND-EGF in the cytosolic domain of A549 cells (Appendix A). This observation supports the statement that FND is only effective when coated with EGF leading to FND-EGF uptake to A549 cells.

In Figure 5, the MFC analysis of cells (*n* = 50) treated with FNDs showed no change in the MFC from 2.1 ± 0.1 (time point 0 min) to 2.0 ± 0.1 (time point 50 min) (*p* = 0.8). The MFC of cells (*n* = 50) treated with free EGF showed an increase in MFC from 2.0 ± 0.1 (time point 0 min) to 2.7 ± 0.1 (time point 50 min) (*** *p* < 0.01). The MFC of cells (*n* = 50) treated with 100 nM FND for 22 h then later treated with conjugated 100 nM FND-EGF showed a higher increase in MFC from 2.1 ± 0.1 (time point 0 min) to 2.8 ± 0.2 (time point 50 min) (*** *p* < 0.01). This finding shows that both free EGF and FND-EGF will lead to mitochondrial fragmentation; namely, the presence of EGF will alter mitochondrial fission. In addition, an increase in FND concentration inside the cytosol domain may induce cell stress and the corresponding mitochondrial dynamics and therefore lead to the highest increase in mitochondrial fragmentation, as shown in Figure 5.

### 3.4. A549 Cells Treated with 1 µM PD153035, 100 nM Cetuximab and 1 mM Caffeine in the Presence of 100 nM Conjugated FND-EGF and Then Exposed to nNIR

FND alone treated with A549 cells did not affect MFC as demonstrated in this study from Figure 5. We further use FND alone to investigate the mitochondrial pattern in the absence of EGF stimulation in response to the tyrosine kinase inhibitor PD153035. When A549 cells were first treated with 1 µM PD153035 and 100 nM FND, followed by 224.02 J/cm^2^ nNIR, the results (Figure 6A) showed that the MFC value sharply increased from 2.1 ± 0.3 to 3.4 ± 0.3 (*p* = 0.003) within 10 min of nNIR irradiation, but the MFC value gradually decreases from 3.4 ± 0.3 to 2.8 ± 0.3 with the increase of time (black line). However, when A549 cells were treated with only 1 µM PD153035 and 100 nM FND, the MFC value decreased with time in the first 10 min, where MFC varied from 1.8 ± 0.3 to 1.2 ± 0.3 (*p* = 0.16), and the MFC tended to level off with the increase of time from 1.2 ± 0.3 to 1.3 ± 0.3 (brown line). Therefore, when A549 cells were treated with FND alone, the MFC = 2.1 ± 0.1 but when cells were treated with FND and PD153035 the MFC = 1.5 ± 0.3, which demonstrated that PD153035 promotes mitochondrial fusion.

We then evaluated the effect of nNIR on A549 cells pretreated with cetuximab (100 nM) and FND-EGF (100 nM) and with caffeine (1 mM) and 100 nM FND-EGF (100 nM). All A549 cells treated with drugs such as PD153035, cetuximab, and caffeine combined with FND-EGF all showed a decrease in MFCs, where a decrease in MFC for A549 cells treated with PD153035, cetuximab, and caffeine combined with FND-EGF from 2.1 ± 0.3 to 0.5 ± 0.3, 1.8 ± 0.3 to 0.6 ± 0.3 and 1.9 ± 0.3 to 0.7 ± 0.3, respectively, as can be seen in Figure 6B–D. However, there was an increase in MFC for A549 cells treated FND-EGF from 2.1 ± 0.1 (time point 0 min) to 2.8 ± 0.2 (time point 50 min). Therefore, in the absence of nNIR irradiation, an increase in MFC due to the presence of FND-EGF will be inhibited by PD153035, cetuximab, and caffeine; these findings suggest these three drugs promote mitochondrial fusion.

### 3.5. ComparativeAanalysis of A549 Cells Treated with Different Classes of Drugs and Then Exposed to nNIR

Figure 7 summarizes the experimental results from Figure 3 to Figure 6. The MFC analysis demonstrated that the positive control mdivi-1 showed a significant decrease in MFC compared to the control; the negative control CCCP showed an increase in MFC compared to the control (Figure 7). In addition, all cells pretreated with the different classes of drugs (1 µM PD153035, 1 mM caffeine, and 100 nM cetuximab) followed by 224.02 J/cm^2^ nNIR irradiation showed significant decreases in the MFC compared to the cells exposed to nNIR irradiation only; however, the MFC values still more significant than the control, which implied nNIR irradiation of single A549 cells induces mitochondrial fission and an increase in MFC due to the presence of nNIR irradiation could suppress PD153035, caffeine, and cetuximab activities in this study.

## 4. Discussion

In this study, when a single cell nucleus was exposed to NIR, there was a sharp increase in MFC from 10 to 15 min then followed by a gradual increase in MFC after 15 min compared to non-exposed NIR cell (MFC: 3.7 ± 0.2 nNIR cell vs. 1.9 ± 0.2 non-exposed NIR cell) Figure 2B, we further investigated an average of 110 cells exposed to nucleus NIR compared to non-nucleus exposed cells, our data analysis showed that there was a sharp increase in MFC within 5 to 15 min in nucleus exposed cells compared to cytosol exposed cells that showed no change in MFC (MFC: 4.1 ± 0.3 nNIR; *** *p* < 0.01 vs. 2.1 ± 0.3 cNIR) Figure 4D. We have shown in this study that single cell nucleus exposed to NIR leads to an increase in MFC and that 110 cells exposed to nucleus NIR and showed an increase in MFC. In this study, nNIR have shown to increase mitochondria fragmentation which leads to a higher MFC and studies have shown that mitochondrial contained nucleic acid [38], further studies have shown that DNA double-strand breaks (DSBs) is said to occur in the nucleus [39] by exposing cell nucleus to nNIR to cause the fragmentation of mitochondrial directly affect the nuclear DNA and therefore may lead to DNA damage. In summary, nNIR leads to the mitochondrial fragmentation and since mitochondrial contained nucleic acid [38] and that DNA controls the functional properties of nucleic acid by damaging the nucleic acid damages the DNA and since double strand break occurs in the nucleus and it is said to be a cytotoxic lesion nNIR may have enhance DNA damage and even double strand break (DSBs) [39]. Further studies are needed to use Phospho-Histone H2A.X (Ser139) (20E3) Rabbit mAb (Alexa Fluor^®^ 647 Conjugate) to demonstrate nucleus DNA damage. This dye can be used on cells that have been exposed to NIR light that may cause DNA damage by presenting a fragmented DNA in the nucleus.

Cells treated with free EGF showed a moderate increase in MFC (MFC = 2.7 ± 0.1) on the other hand, cells treated with FND only showed no increase in MFC (MFC = 2.1 ± 0.1) and there was a small amount of non-specific uptake of FND in the cytosol of the cell (Appendix A). These data have shown that due to the synergetic effects of increase concentration of FND and EGF tend to have influence an increase in MFC since studies have shown that a higher concentration of FND may lead to oxidative stress in the balance of energy metabolism and dysfunction of mitochondrial [40]. These two factors, that is the increased concentration of FND and the 100 nM EGF. may have influenced the increase in MFC (Figure 5). In order for us to determine the FND and EGF activity, A549 cells were treated with 100 nM FND and incubated for 22 h we then treat these cells with conjugated 100 nM FND-EGF another batch of cells were treated with 100 nM free EGF and another set of cells were treated with FND only, our data analysis showed that cells treated with conjugated FND-EGF presented a high MFC (2.8 ± 0.2), and the fluorescence images shows very large amount of FND in the cytosolic domain of A549 cell (Appendix A).

We further treated cells with PD153035 and FND and exposed the nucleus of these cells to 224.02 J/cm^2^ NIR and treated another set of cells with PD153035 and FND only. Analysis showed a sharp increase in MFC of PD153035 exposed cells to NIR from 0 to 10 min (MFC from 2.1 ± 0.3 to 3.4 ± 0.3). However, there was a gradual decrease in the MFC from 3.4 ± 0.3 to 2.8 ± 0.3 (Figure 6A). Our data showed that cells treated with PD153035 and FND only showed a sharp decrease in MFC from 0 to 10 min MFC from 1.8 ± 0.3 to 1.2 ± 0.3 but was stable after 10 min (MFC 1.2 ± 0.3 to 1.3 ± 0.3), as shown in Figure 6A. On the other hand, this result shows that in the absence of nNIR, PD153035 will inhibit the signal pathway of EGF, and studies have shown that it is a potent inhibitor of EGF tyrosine kinase. However, it thus also binds to the DNA double-helical structures by insertion [19]; this may therefore lead to mitochondrial fusion, but on the other hand, in the presence of nNIR, PD153035 function may be suppressed, which may lead to DNA damage and that FND alone is not sufficient enough to suppress PD153035 activity (Figure 6A). However, when A549 cells were treated with 1 µM PD153035 and 100 nM FND-EGF followed by 224.02 J/cm^2^ nNIR, the results showed a sharp increase in MFC from 2.1 ± 0.3 to 3.7 ± 0.3 but gradually decreased from 3.7 ± 0.3 to 2.8 ± 0.3 with time in the first 10 min, but the MFC tended to level off with the increase of time (Figure 6B).

A total of 100 nM of Cetuximab was treated with conjugated 100 nM FND-EGF and the cell nucleus was exposed to NIR. Analysis showed a gentle increase in MFC from 10 to 20 min but was stable after 20 min. On the other hand, cells were treated with conjugated 100 nM FND-EGF and100 nM Cetuximab, and data analysis showed that there was a stable MFC from 0 to 20 min but a sharp decrease in MFC from 20 to 40 min then it maintained stability MFC (MFC: 3.2 ± 0.3 nNIR, vs. 0.8 ± 0.3 non-exposed NIR), as shown in Figure 6C. Note that when cells are exposed to nNIR in the presence of the cetuximab, the nNIR thus suppresses the cetuximab inhibitory effect. On the other hand, when cells are not exposed to nNIR, cetuximab thus inhibits EGF even in the presence of FND. This observation demonstrated that in the presence of nNIR, the inhibitor had been suppressed, resulting in an increase in MFC due to the nNIR DNA damage effect (Figure 6C).

Cells were also treated with 1 mM caffeine, and the cell nucleus was exposed to NIR. Analysis showed an increase in MFC. On the other hand, cells treated with caffeine and conjugated FND-EGF only showed a decrease in MFC (MFC: 3.3 ± 0.3 nNIR cell, vs. 0.7 ± 0.3 non-exposed NIR cell), as shown in Figure 6D. These results showed that nNIR effect thus leads to an increase in MFC due to the DNA damage effect even in the presence of caffeine.

These MFC’s were compared to a positive and negative control such as mdivi-1 and CCCP. Mdivi-1 inhibited mitochondrial fission [41] and CCCP mediated mitochondrial fission and mitophagy [42]. Using these two controls in our study, we were able to screen PD153035, cetuximab, and caffeine for their role in EGF inhibition in cancer treatment. Using these positive and negative control, we determined that all MFC values in our study above the control but below or equal to the CCCP MFC value were categorized as an increased MFC. This increase in MFC, therefore, leads to mitochondrial fission. On the other hand, all MFC values below the control but above the mdivi-1 MFC value were categorized as a decreased MFC. This decrease in MFC, therefore, leads to mitochondrial fusion. On the other hand, all MFCs in our study that were below the control but above the mdivi-1 MFCs values have been categorized as low MFC, a low MFC therefore leads to mitochondrial fusion. In our study using single cell nuclear NIR light and EGF in combination with treated drugs showed a significant increase in MFC compared to non-nuclear NIR exposed cells, and a significant decrease in MFC compared to only nNIR exposed cells (Figure 6). An increase in EGFR activities may lead to an increase in MFC. Cetuximab that can block EGFR activities [43,44] was used in this study since it blocks EGFR activities, as supported in Figure 6C. But our result also showed that nNIR and FND-EGF may have suppresses cetuximab functions of EGFR inhibition. Caffeine was used in this study since it inhibits the enzyme phosphodiesterases and the calcium channel. This will, as a result, inhibit the DNA damage [45]. However, caffeine function may have been overridden in inhibiting DNA damage due to the presence of nNIR and FND-EGF. FND have been used in intracellular environments of different types of cells [46,47] where they showed a very good biocompatibility [48]. Targeting nondiamond has previously been used in the diagnostics application in cancer and tumors [49]. It is also useful in the sense that they are very visible in different imaging techniques. However, an increase in FND concentration inside the cytosol domain may induce cell stress and the corresponding mitochondrial dynamics and therefore lead to mitochondrial dysfunction.

## 5. Conclusions

In this study, we demonstrated that when a single cell nucleus is exposed to NIR, this will increase MFC. Furthermore, this increase in MFC is associated with increased mitochondrial fission, resulting in mitochondrial fragmentation. Our study illustrated four unique findings. Firstly, single cell nNIR in the same image plane led to an increase in mitochondrial fragmentation compared to non-nuclear exposed single cell. Secondly, single cell nNIR showed a significant increase in MFC compared to cytosol single cell NIR. It is therefore worth noting that NIR have been used for the first time here to distinguish between two subcellular structure such as the nucleus and cytosol to demonstrate that nNIR leads an increase in MFC and to mitochondrial fission caused by the nNIR effect on DNA damage. Thirdly, we monitored the activities of EGF in A549 cell. This study has demonstrated that treating cells with conjugated FND-EGF have a significant increase effect on the MFC compared to either free EGF treated cells and cells treated with FND only and that we have shown that increased concentration of FND may induce cell stress, mitochondrial dysfunction caused by higher concentration of FND [30]. This have demonstrated the importance and limitations of a carrier such as a nanomaterial FND to enhance endocytosis of EGF into the cytosol domain of a cell to be able to determine EGF activities and to provide a better target for drug treatment but FND into the cytosol domain of a cell may induce cell stress, mitochondrial fragmentation, and mitochondrial dysfunction when FND in higher concentration. Fourthly, since EGF is of clinical importance in cancer therapeutics, we targeted EGF activities in the presence of different drugs and exposed cells to nNIR. Our data analysis has shown for the first time a significant increase in MFC’s of cells exposed to nNIR. In this study, we targeted nNIR to demonstrate for the first time that nNIR leads to an increase in MFC and since mitochondrial contained nucleic acid [28] and the DNA controls the functional properties of nucleic acid and the DNA strand break is said to occur in the nucleus [29] we therefore concluded that the increase in MFC may have been caused by nNIR which leads to DNA damage and therefore resulted to mitochondrial fission. A comparative analysis of nNIR treated cells only showed that there was a significant increase in MFC compared to drug treated cells and to the controls cells but that all drugs treated cells exposed to nNIR showed a significant increase in MFC compared to the control and mdivi-1 the positive control this highlight that even in the presence of inhibitor such as cetuximab, PD153035 or caffeine nNIR suppresses these drug functions (Figure 7). In summary, the present study provided a fundamental important information about single cell exposure to nuclear NIR that will lead to the alteration of mitochondria dynamics which subsequently leads to an increase in MFC and therefore leads to mitochondrial fission but not cytosol exposed cells to NIR and that all cells treated with drugs but not exposed to nucleus NIR were all determined as been caused by mitochondrial fusion because of the drug EGF inhibitory effects. This study has demonstrated that all of these drugs when treated alone on cell did not prove to be effective against nNIR and all of these drugs are being liable to resistibility. We anticipate mitochondrial patterns can be used as biomarkers of cancer diagnosis and cancer drug response. In addition, the proposed single-cell method could be applied to construct a rapid screening method for the detection and therapeutic evaluation of many types of cancer.

## Figures and Tables

**Figure 1 cells-11-00624-f001:**
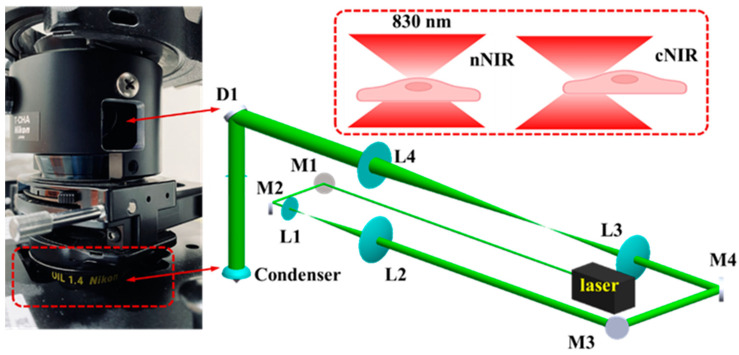
The experimental setup for the single-cell NIR laser irradiation system consists of an 830 nm infrared diode laser, an electrical shutter, laser-focusing optics (condenser), and a specimen holder attaching to the XY-axis motorized stage. The laser beam is expanded eightfold with lens pairs from the combination of 1:4 telescope (L1:L2) and 1:2 telescope (L3:L4) to slightly fill the back aperture of the condenser, where the dichroic mirror D1 is placed above the condenser to reflect the laser beam into the condenser while transmitting visible light for bright-field imaging. The upper right is the two operation modes for the single-cell NIR laser irradiation: the laser focal spot is located either in the nucleus (nNIR) or in the cytosol (cNIR) of single cell.

**Figure 2 cells-11-00624-f002:**
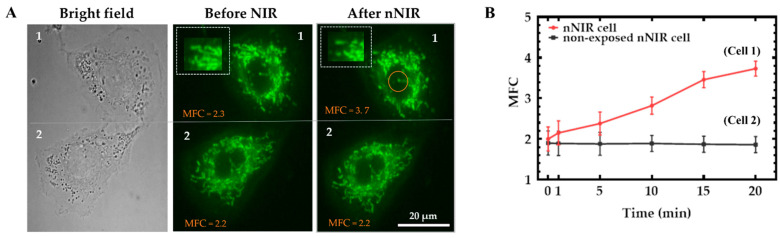
Exposure of A549 cell before and after to nuclear 224.02 J/cm^2^ NIR for 10 s. (**A**) Cell 1 on the left panel illustrates the bright field of the cell. Cell 1 located in the middle panel is the fluorescence image before nNIR, and cell 1 on the top right panel illustrates cell 1 after 20 min of nNIR (laser spot area in brown circle cell 1), where the fluorescence imaged cell showed a fragmented mitochondrial structure in cell 1 located at the top right panel after 20 min of nNIR. (**B**) MFC of single cell nuclear NIR showed a significant (*p* < 0.01) increase of nuclear exposed cell (Cell 1: MFC = 3.7 ± 0.2) but no MFC change in non-exposed nNIR cell (Cell 2: MFC = 1.9 ± 0.2).

**Figure 3 cells-11-00624-f003:**
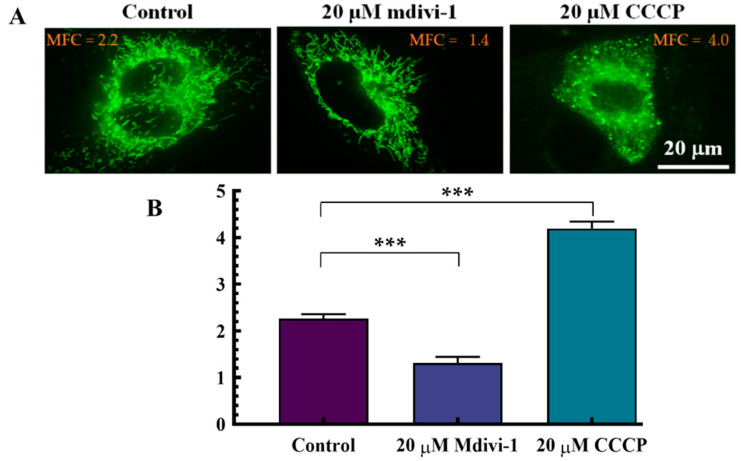
Cells treated with 20 µM mdivi-1 and 20 µM CCCP. (**A**) illustrates cells treated with 20 µM mdivi-1 and 20 µM CCCP and incubated for 30 min. There was a well elongated and distinct mitochondrial structure shown by cells treated with mdivi-1 but a fragmented and granular mitochondrial structure shown for CCCP treated cells compared to the control. There was a significant (*** *p* < 0.01) decrease in MFC (1.5 ± 0.2) of mdivi-1 treated cells compared control, but CCCP treated cells showed a very significant (*** *p* < 0.01) increase in MFC (4.3 ± 0.2) compared to the control shown in (**B**).

**Figure 4 cells-11-00624-f004:**
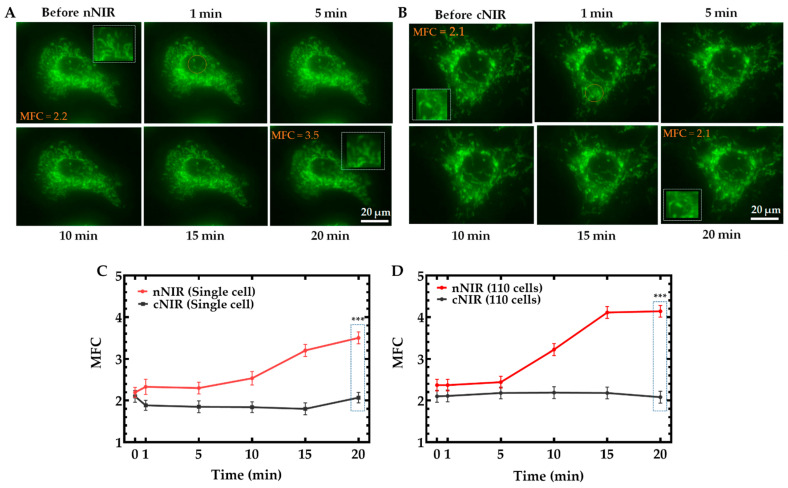
A549 cells (*n* = 50) exposed to 224.02 J/cm^2^ nNIR and cNIR for 10 s and imaged in a time-dependent manner at 0, 1, 5, 10, 15, and 20 min. (**A**) Mitochondrial dynamics of a single cell exposed to 224.02 J/cm^2^ nNIR showed significant mitochondrial structural fragmentation. (**B**) Mitochondrial dynamics of a single cell cytosol exposure to 224.02 J/cm^2^ cNIR, and an image analysis showed no significant impact on mitochondrial dynamics. (**C**) The MFC of a single A549 cell exposed to (**A**) nNIR and (**B**) cNIR showed a very significant (*** *p* < 0.01) increase in MFC of nNIR compared to cNIR (MFC: 3.50 ± 0.2 nNIR; *** *p* < 0.01 vs. 2.1 ± 0.2 cNIR). (**D**) An average of 110 single cells exposed to nNIR and cNIR (MFC: 4.1 ± 0.3 nNIR, vs. 2.1 ± 0.3 cNIR) were compared, and there was a very significant (*** *p* < 0.01) difference in MFCs after 5 min of NIR exposure between the nNIR- and cNIR-exposed cells. Statistical analysis was performed by a one-way ANOVA.

**Figure 5 cells-11-00624-f005:**
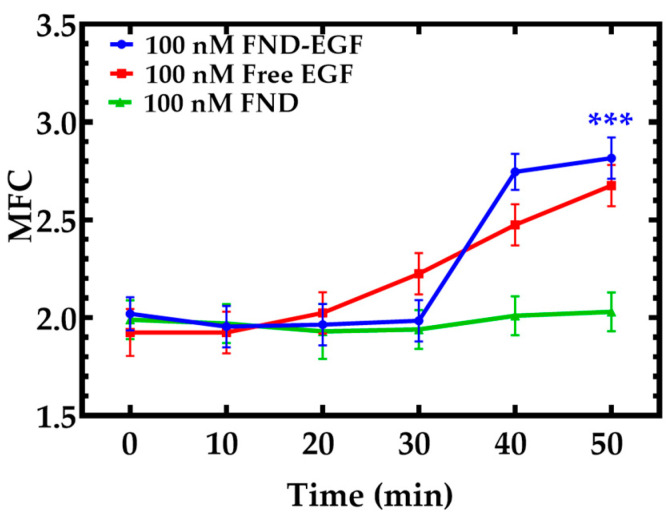
The MFC analysis of A549 cells treated with FNDs, free EGF, and FND-EGF. MFC for cells treated with free EGF showed a moderate increase in MFC (MFC = 2.7 ± 0.1) but there was no change in MFC for cells treated with FND only (MFC = 2.1 ± 0.1). The MFC of cells treated with 100 nM FND for 22 h then later treated with conjugated 100 nM FND-EGF showed a very significant (*** *p* < 0.01) increase in MFC (2.8 ± 0.2).

**Figure 6 cells-11-00624-f006:**
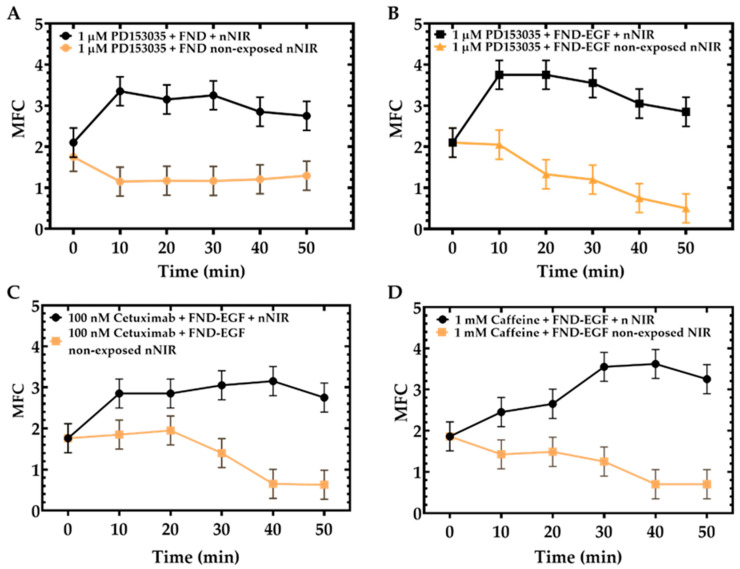
Drug treatment with A549 cells (*n* = 50) and exposed to 224.02 J/cm^2^ nNIR. (**A**) illustrates cells treated with 1 µM PD153035 and 100 nM FND was exposed to 224.02 J/cm^2^ nNIR for 10 s there was an increase in MFC compared to non-nuclear exposed cell (MFC: 2.8 ± 0.3 nNIR, vs. 1.3 ± 0.3 non-NIR). (**B**) illustrates cells treated with 1 µM PD153035 and conjugated 100 nM FND-EGF then exposed to 224.02 J/cm^2^ nNIR for 10 s and imaged at different time points, analysis showed an increase in MFC (MFC: 2.9 ± 0.3 nNIR, vs. 0.5 ± 0.3 non-NIR) of cell nucleus exposed to NIR compared to non-nuclear exposed cells. Cetuximab treated A549 cells and conjugated 100 nM FND-EGF were exposed to 224.02 J/cm^2^ nNIR, and analysis showed an increase in MFC compared to the non-nuclear exposed cells (MFC: 2.8 ± 0.3 nNIR vs. 0.6 ± 0.3 non-NIR), as shown in (**C**). A549 cells treated with conjugated 100 nM FND-EGF and 1 mM caffeine were exposed to 224.02 J/cm^2^ nNIR at different time points. Analysis showed an increase in MFC of nuclear exposed cells compared to non-NIR cells (MFC: 3.3 ± 0.3 nNIR, vs. 0.7 ± 0.3 non-NIR), as shown in (**D**). Statistical analysis was performed by one-way ANOVA.

**Figure 7 cells-11-00624-f007:**
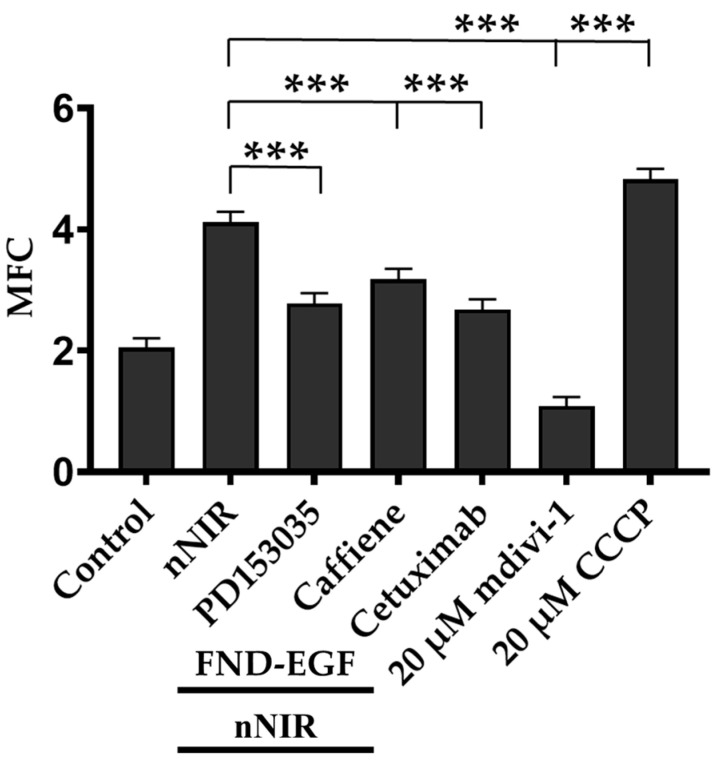
Comparative analysis of A549 cells exposed to nNIR and A549 cells treated with different classes of drugs (1 µM PD153035 and 100 nM FND/1 mM caffeine and 100 nM FND/100 nM Cetuximab and 100 nM FND) followed by exposure to nNIR. There was a significant *** *p* < 0.01 increase MFC of nNIR compared to drug treated cells (MFC: 4.1 ± 0.3 nNIR cell vs. 2.9 ± 0.3 PD153035 treated cells, vs. 3.3 ± 0.3 caffeine treated cells vs. 2.8 ± 0.3 cetuximab treated cells). There was a significant *** *p* < 0.01 increase in MFC of nNIR compared to mdivi-1 (MFC: 4.1 ± 0.3 nNIR vs. 1.5 ± 0.2 mdivi-1 treated cells). On the other hand, there was a significant *** *p* < 0.01 increase in MFC of CCCP treated cells compared to nNIR cell (4.3 ± 0.2 CCCP treated cells vs. 4.1± 0.3 nNIR).

## Data Availability

Raw image data and analyzed data can be found in the attached folder and these data were generated during the study.

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
