# Peer review of "Nucleus Near-Infrared (nNIR) Irradiation of Single A549 Cells Induces DNA Damage and Activates EGFR Leading to Mitochondrial Fission"

_cells, 2022, doi:10.3390/cells11040624_

Round 1

Reviewer 1 Report

In this manuscript, the authors reported a near-infrared laser focus into nucleus could regulate the mitochondrial dynamics of cancer cells. It was demonstrated that such a light regulated biological activity was similar with Cetuximab, PD153035, et al. Overall, this study provided a novel insight of the light controlled cell activities. Some specific comments were listed as bellow:

  1. A brief introduction of the laser used in this study was suggested to be described. How to distinguish nucleus and cytoplasm during the NIR irradiation?
  2. NIR mediated DNA damaging should be confirmed.
  3. No significant difference was observed in Figure 2A, Figure 4A and Figure 4B.
  4. The methods and the replicates of the quantitative analysis should be provided.

Author Response

The authors would like to thank Reviewer #1 for the careful review of our manuscript and for providing us with constructive comments and suggestions to improve the quality of the manuscript. Your concern has been carefully addressed, as can be seen below. Modifications are highlighted by red color words in the markedly revised manuscript to help the Reviewer and the Editor for an easier checking on the changes we made.

Reviewer 2 Report

          The work of Gbetuwa and colleagues is devoted to the study of the novel role of cell nucleus in mediating near-infrared irradiation (NIR) effects on mitochondrial dynamics of A549 cancer cells. It is very important to understand the biological basis of cellular response to NIR in order to improve the NIR phototherapy and avoid over-dosing related complications. The authors expect that mitochondrial patterns can be used as biomarkers of cancer diagnosis and cancer drug response and the proposed single-cell method could be applied to construct a rapid screening method for the detection and therapeutic evaluation of many types of cancer.           The manuscript may be of interest to a wide range of readers, but the text needs careful checking and revision.           The main question that I would like to address to the authors is regarding the applied radiation flux density. Why was the value of 224.02 J /cm2 chosen and how does it correlate with the radiation flux density during phototherapy in the clinic? It would be great to see explanations in the text.          In addition, the text often lacks punctuation marks (for example, line 72 after “and physical properties” missing point, line 85 after “cytosol NIR (cNIR)“ missing point, line 91 after “on A549” missing comma, line 225 after “50 single cells” missing point). There are entire paragraphs in which there is a noticeable inconsistency of times (especially in the section materials and methods, for example, sections 2.5, 2.6).            There are many duplicate words in the text (line 63 “ability to be able”, line 74 “a range of applications that ranges”, line 318 “probe for probing”), which can be replaced with synonyms.              All this complicates the reading and perception of the text.        It is also worth noting a number of typos and shortcomings: line 78 “FND” occurs for the first time in the text and is not deciphered;
line 61 “in vitro and in vivo” should be italicized; line 75 “in vivo” should be italicized; line 88 “oA549” it was meant “of A549”? line 102 “Thermos Fisher” should be “Thermo Fisher”; line 111 "37 °C" invalid character; line 163 “with A549 cell” should be “of A549 cells”; lines 163, 171, 186, 197 it is more precisely to speak “treatment of A549 cells” instead of “treatment of A549 cell”; lines 184, 193, 204 “224.02J/cm2”; line 210 “imageJ” missing spaces; line 193, 204 “expose” should be “Expose”; lines 225 should be “50 single cells”; lines 269, 285; 33; 417, 436 problems with figures layout; line 316 “Free EGF, and” extra comma; lines 357, 364, 476, 488 “224.02 J/cm2” should be “224.02 J/cm2”; line 477 “PD163035” should be “PD153035”; line 508 “mdivi-1.” Duplicated; line 567 “bur not” should be “but not”.
     In general, after revision, the manuscript can be recommended for publication in the journal "Cells".

Author Response

The authors wish to thank Reviewer #2 for the careful review of our manuscript and for providing us with constructive comments and suggestions for the manuscript to be carefully checked and revised to better improve the manuscript. Your concerns have been carefully addressed and can be seen below. Modifications are highlighted by red color words in the markedly revised manuscript to help the Reviewer and the Editor for an easier checking on the changes we made.

Reviewer 3 Report

This manuscript reported a study that examined the effect of nuclear NIR irradiation on EGFR activities in cells treated with various drugs, with potential application in cancer treatment. The information is of interest. However, the presentation of the manuscript should be significantly improved.

Major Concerns:

  1. The measurements in the control experiments lasted 20 minutes (Figs. 2 and 4), while the measurements in the other experiments lasted 50 minutes (Figs. 5 and 6). The results of the experiments cannot be compared to the results of the control experiments because significant changes occurred after the first 20 minutes in some of the long experiments. The control experiments should be performed for 50 minutes, like the other experiments.
  2. Some of the graphs' error bars, particularly in Figure 5, are incorrect and misleading.
  3. The claim that the FND is located in the cell membrane or cytosol is not supported by the figures in the Supplementary data. From Fig. s1 you cannot conclude that: "There were scattered FND around the cell membrane but not in the cytosol of the cell"
  4. After demonstrating that FND alone had no effect (Fig. 5), it is unclear why the authors chose to use FND alone with the drug treatment (Fig. 6A).

Minor Concerns:

  • Abbreviations are given without their interpretation: line 77 (ND), line 78 (FND), line 89 (MFC).
  • Section 2.5 in Materials and Methods: It was not reported how long after exposure to NIR the measurements were performed.
  • Unnecessary repetition. It is advisable to combine sections 2.6-2.8 in Materials and Methods.
  • Line 225: Rewrite the sentence.
  • Figure 2: The graph (2B) is redundant. There is no justification for presenting a single measurement data, especially since the statistics of 110 measurements is shown in Figure 4D. Correct the errors bars in 2B.
  • Figure 4: The graph (4C) is redundant. There is no justification for presenting a single measurement data, especially since the statistics of 110 measurements is shown in Figure 4D. Correct the errors bars in 2B.
  • Line 327: You cannot claim for "highly significant increase". There is an overlap between the results and therefore no significance can be argued
  • Line 330: There are no supporting evidence for this claim.
  • Figure 5: Error bars are incorrect and misleading.
  • Line 365: Not true.
  • Figure 6: It is not clear why the graph 6A is needed. Fix the error bars.
  • Line 389: The information does not match the graph.
  • Line 394: "we then… FND-EGF" should be deleted.
  • Line 441: Discussion of the results of the individual measurement is unnecessary.
  • Line 447: The numbers provided are incorrect.
  • Line 462: You cannot claim that: " the FND were found scattered around the cell membrane but not into the cytosolic 462 domain" the Figure S1 does not support that.
  • Line 475: After demonstrating that FND alone had no effect, it is unclear why the authors chose to use FND alone with the drug treatment.
  • Line 502: Should add FND-EGF in the beginning of the sentence.
  • Line 507: Should be caffeine instead of cetuximab.
  • Line 521: Rewrite the sentence.
  • Line 534: Rewrite the sentence.
  • Line 543: Not true.
  • Line 558: Need more evidence to support this claim.

Author Response

The authors would like to thank Reviewer #3 for his or her careful review on our manuscript and for providing us with constructive comments and suggestions to further improve the quality of our manuscript. Your concern has been carefully addressed by the following point-by-point responses. Modifications are highlighted by red color words in the markedly revised manuscript to help the Editor and Reviewer #3 for an easier checking on the changes we made.

Round 2

Reviewer 1 Report

The authors have addressed most of my concerns, and I have no more questions.